# Cell–Matrix Interactions in the Eye: From Cornea to Choroid

**DOI:** 10.3390/cells10030687

**Published:** 2021-03-20

**Authors:** Andrew E. Pouw, Mark A. Greiner, Razek G. Coussa, Chunhua Jiao, Ian C. Han, Jessica M. Skeie, John H. Fingert, Robert F. Mullins, Elliott H. Sohn

**Affiliations:** 1Department of Ophthalmology and Visual Sciences, Carver College of Medicine, University of Iowa Hospitals & Clinics, Iowa City, IA 52242, USA; andrew-pouw@uiowa.edu (A.E.P.); mark-greiner@uiowa.edu (M.A.G.); razekgeorges-coussa@uiowa.edu (R.G.C.); chunhua-jiao@uiowa.edu (C.J.); ian-han@uiowa.edu (I.C.H.); jessica-skeie@uiowa.edu (J.M.S.); john-fingert@uiowa.edu (J.H.F.); robert-mullins@uiowa.edu (R.F.M.); 2Institute for Vision Research, University of Iowa, Iowa City, IA 52242, USA

**Keywords:** TGF-beta, descemet membrane, collagen, metalloproteinases, AMD, interphotoreceptor matrix, MMP-9, VEGF, MMP-14, TIMP-3, bruch’s membrane, choroid

## Abstract

The extracellular matrix (ECM) plays a crucial role in all parts of the eye, from maintaining clarity and hydration of the cornea and vitreous to regulating angiogenesis, intraocular pressure maintenance, and vascular signaling. This review focuses on the interactions of the ECM for homeostasis of normal physiologic functions of the cornea, vitreous, retina, retinal pigment epithelium, Bruch’s membrane, and choroid as well as trabecular meshwork, optic nerve, conjunctiva and tenon’s layer as it relates to glaucoma. A variety of pathways and key factors related to ECM in the eye are discussed, including but not limited to those related to transforming growth factor-β, vascular endothelial growth factor, basic-fibroblastic growth factor, connective tissue growth factor, matrix metalloproteinases (including MMP-2 and MMP-9, and MMP-14), collagen IV, fibronectin, elastin, canonical signaling, integrins, and endothelial morphogenesis consistent of cellular activation-tubulogenesis and cellular differentiation-stabilization. Alterations contributing to disease states such as wound healing, diabetes-related complications, Fuchs endothelial corneal dystrophy, angiogenesis, fibrosis, age-related macular degeneration, retinal detachment, and posteriorly inserted vitreous base are also reviewed.

## 1. Introduction

The extracellular matrix (ECM) is an essential and major component of the ocular microenvironment. It forms a complex but organized meshwork surrounding cells and confers not only cellular structural and mechanical support, but also regulates cellular homeostasis and signaling [1]. Proteoglycans (including heparan sulfate, chondroitin sulfate, and keratin sulfate), hyaluronic acid, collagen, elastin, laminin, fibronectin, and fibrillin represent major components of the ECM [2,3]. Other components include extracellular proteases (such as matrix metalloproteinases, aka MMPs), immune mediators and growth factors [4].

The first portion of this manuscript is a review loosely organized from the front to the back of the eye, starting with the cornea, then addressing parts of the eye involved in intraocular pressure maintenance and glaucoma (i.e., trabecular meshwork, optic nerve, conjunctiva and tenon’s layer), vitreous, retina, retinal pigment epithelium, Bruch’s membrane, and choroid. Each section describes how ECM components are involved in homeostasis but also details its alterations resulting in disease states such as wound healing, diabetes-related complications, Fuchs endothelial corneal dystrophy, angiogenesis, fibrosis, age-related macular degeneration, choroidal neovascularization, retinal detachment, and posteriorly inserted vitreous base.

## 2. Cornea

Under normal conditions, the cornea is able to support visual transparency by remaining avascular. A complex counterbalance of homeostatic mechanisms exist to maintain corneal avascularity. In the most anterior layers of the cornea, these mechanisms include soluble factors in the precorneal tear film that mediate corneal immune privilege, such as transforming growth factor-*β* (TGF-*β*); the limbal stem cell niche, which prevents conjunctivalization of the epithelium; and structural factors in the stroma that prevent vascular ingrowth mechanically, such as tight packing of the collagen lamellae and dense corneal innervation [5]. Perturbations of these homeostatic conditions, such as those occurring with trauma, aging and various infectious and inflammatory diseases, can result in degeneration of these functional barriers against opacification due to vascular infiltration and fibrosis of the cornea [6]. Corneal opacification due to scarring and vascularization may require optical rehabilitation with contact lenses or surgical rehabilitation with keratoplasty and can lead to blindness.

In addition to corneal avascularity, the cornea under normal conditions is able to support visual clarity by maintaining appropriate hydration. The corneal endothelium-Descemet membrane (EDM) complex—the functional unit that comprises the two innermost layers of the cornea—is the primary regulator of corneal hydration. Within this complex, the corneal endothelial cell (CEC) monolayer functions as the primary boundary between the corneal stroma and the anterior chamber. CECs maintain the clarity of the cornea by regulating corneal stroma hydration through barrier and pump functions. However, the EDM undergoes changes with aging. There is a high density of endothelial cells at birth, but these cells are arrested in G1 phase and therefore do not repopulate themselves following death of adjacent cells. As a result, human corneal endothelial cell density decreases at an average rate of approximately 0.6% per year in normal corneas throughout adult life, with gradual increases in polymegathism and pleomorphism to maintain a continuous endothelial cell layer [7]. Descemet membrane, the basement membrane of the corneal endothelium, also undergoes age-related changes. This specialized extracellular matrix is comprised of a fetal anterior banded layer that is present at birth and a posterior non-banded layer that thickens with age as CECs continue to secrete extracellular matrix proteins. Reductions in endothelial cell density and alterations to Descemet membrane occur together in aging and in pathological conditions affecting the posterior cornea. As CECs do not divide readily, sodium-potassium adenosine triphosphatase pump sites and cell-cell junctions also decline when the endothelial cell density falls below a critical level. When the EDM undergoes cell death that exceed age-predicted changes, premature corneal edema and vision loss can occur, and keratoplasty using donor CECs may be indicated.

We will review the effects of damaging conditions on the anterior corneal layers, and the interplay between corneal neovascularization and extracellular matrix alterations after epithelial and stromal wounding. With the wide variety of disease conditions that can lead to the development of corneal neovascularization (e.g., infection, inflammation, trauma, degeneration), it is important to understand the mechanisms by which neovascularization can lead to loss of stromal clarity. We will also review the effects of two common diseases affecting cell–matrix interactions in the EDM complex—Fuchs endothelial corneal dystrophy and diabetes mellitus—and the impact of these pathologies on posterior corneal health and function over time. With the emergence of single-layer endothelial keratoplasty techniques such as Descemet membrane endothelial keratoplasty (DMEK) and Descemet stripping automated endothelial keratoplasty (DSAEK), it is imperative to understand cell–matrix interactions in the posterior cornea in more detail, particularly because posterior lamellar donor tissues may be altered by disease states such as diabetes [8,9,10].

### 2.1. Stromal Extracellular Matrix Alterations and Corneal Neovascularization

The cornea is able to maintain clarity and avascularity, even in most cases after sustaining injury to the epithelium and stroma, by a variety of mechanisms that preserve the homeostatic balance between pro-angiogenic and anti-angiogenic factors. However, corneal neovascularization can occur in a variety of conditions—including microbial keratitis, autoimmune and systemic inflammatory conditions, corneal graft rejection, neurotrophic keratitis, chemical injury, contact lens overwear, and limbal stem cell deficiency—where angiogenesis is initiated despite the presence of homeostatic anti-angiogenic regulatory mechanisms [6]. Most often, when corneal neovascularization does occur, it involves the anterior two-thirds of the stroma (89%) and is frequently associated with corneal edema and/or inflammatory cell recruitment [11].

Corneal stromal wound healing occurs in 4 phases [5]. In the first phase, keratocytes at the area of wounding undergo apoptosis, which initiates a healing response and leaves a central acellular zone [12]. In the second phase, adjacent keratocytes immediately begin to proliferate to repopulate the wound area and transform into fibroblasts that migrate into the wound area, a process that can take days. In the third phase, transformation of fibroblasts into myofibroblasts occurs, and in the fourth phase cell-mediated remodeling of the stroma occurs, which can take more than 1 year. The process of stromal wound healing is mediated by signaling of transforming growth factor-beta (TGF-*β*) [13], matrix metalloproteinases (MMPs), and a balance between pro-angiogenic factors (vascular endothelial growth factor [VEGF], basic fibroblastic growth factor [bFGF], also referred to as FGF-2], and platelet-derived growth factor [PDGF]) and anti-angiogenic factors (angiostatin, endostatin, pigment epithelium-derived factor [PEDF], thrombospondin-1, and soluble VEGF receptor 1 [VEGFR1]) [5,14] (Figure 1).

When the balance between pro-angiogenesis and anti-angiogenesis is not maintained, corneal neovascularization can occur (Figure 2). As a result of corneal epithelial and stromal injury, bFGF becomes upregulated and mediates fibroblast activation, whereas stromal fibroblast MMP-14 initiates enzymatic activity [5]. bFGF-mediated fibroblasts and stromal fibroblasts show upregulation of VEGF, and MMP-14 potentiates bFGF-induced corneal NV [15]. In addition, MMP-14 also mediates the degradation of ECM. Both VEGF upregulation and ECM degradation enhance vascular endothelial cell proliferation, migration, and tube formation. In addition to MMP-14 and VEGF, vascular growth in the corneal stroma is also associated with MMP-2, tissue inhibitors of metalloproteinases-2 (TIMP-2), and Src [5,16], and requires both cell proliferation/migration and extracellular matrix turnover.

Of note, MMP-14 is the most prevalent MMP involved in angiogenesis and ECM remodeling in the cornea [17,18], and induces angiogenesis and ECM degradation by a variety of other signaling pathways in addition to the bFGF pathway [5]. MMP-14 activity leads to the disruption of endothelial tight junctions, reorganization of the actin cytoskeleton, and proteolysis of the basement membrane and interstitial matrix. Furthermore, MMP-14 cleaves ECM molecules such as type I collagen, degrading the ECM as well as stimulating migration, organization, and guidance of vascular endothelial cells to form new blood vessels [19].

### 2.2. Fuchs Endothelial Corneal Dystrophy (FECD) and EDM Pathology

FECD, the most prevalent indication for endothelial keratoplasty in the United States [20], results in both CEC dysfunction and abnormal extracellular matrix deposition that are observable clinically as corneal edema and characteristic excrescences on the posterior cornea (guttae) [19]. Endothelial cell characteristics of FECD include channel protein dysfunction, mitochondrial dysfunction, reactive oxygen species accumulation, endoplasmic reticulum stress, DNA alterations, unfolded protein response, and cell apoptosis and dropout [21,22].

Experimental data using tissues from patients with FECD as well as expanded, immortalized cell cultures that model FECD demonstrate the overexpression of collagen 1, fibronectin, and collagen 4 in the EDM complex [23]. These protein expression changes are the result of zinc finger E-box binding homeobox 1 (ZEB1) and Snail1 activation in FECD, which can be stimulated and/or augmented with TGF-*β* exposure. The result of this disease-mediated overexpression is both thickening of Descemet membrane and buildup of matrix proteins that result in corneal guttae formation.

Corneal guttae indicate regions of endothelial cell dropout in FECD. Until recently, it was not clear whether extracellular matrix buildup caused cellular dropout or whether cell loss resulted in matrix changes and guttae formation. Importantly, Kocaba et al. have tested the impact that Descemet membrane tissues with guttae have on normal corneal endothelial cell health. After seeding healthy immortalized CECs on decellularized Descemet membrane explants from FECD patients undergoing surgery, they observed a decrease in the coverage area and number of normal cells seeded onto these abnormal membranes, and an increase in apoptosis surrounding large diameter guttae, compared to decellularized membranes from healthy guttae-free control donor corneas [24]. The same group also noted an increase in the expression of alpha-smooth muscle actin, N-cadherin, Snail1, and NOX4 in normal CECs that were grown in the presence of large diameter guttae. These expression footprints indicate that endothelial-mesenchymal transition (EndoMT) with loss of cell phenotype, as well as increased extracellular matrix component transcription, occur in conjunction with apoptotic cell death in FECD. Taken together, these findings indicate that the presence of extracellular matrix alterations alone are capable of causing altered expression of EndoMT proteins and additional matrix proteins, yielding a vicious cycle of aberrant cell and matrix changes that over time can result in Descemet membrane thickening, guttae formation, and ultimately endothelial cell phenotype loss and cell death (Figure 3).

### 2.3. Diabetes and EDM Pathology

Type II diabetes mellitus (T2DM) causes dramatic pathological effects on the corneal EDM complex and can result in vision loss due to corneal edema. Characteristic disease-related changes include lower than average CEC densities [25], greater than average postoperative cell loss [9], altered matrix properties altering EDM biomechanical stiffness [24], and reduced mitochondrial quality. Endothelial cell dropout in patients with diabetes may result in greater than average need for keratoplasty. Importantly, changes in both cell health and Descemet membrane that occur in tandem due to diabetes are important indicators for keratoplasty surgical outcomes, particularly as they relate to donor tissue health [8].

T2DM CECs have several functional changes that may explain the greater amount of cell loss observed in donor corneal tissues. As described by Aldrich et al., CECs from donors with a history of advanced diabetes have lower mitochondrial respiratory capacity than control cells, as measured using the Seahorse Extracellular Flux Analyzer [26]. Looking closely at diabetic CEC mitochondrial morphology using transmission electron microscopy, there are increases in cristae dropout, inclusion body formation, and average surface area in donor tissues with T2DM. Taken with the functional data, these findings demonstrate that the mitochondria in CECs with T2DM have a decreased functional capacity despite a larger surface area, and indicate an imbalance in mitochondrial dynamics related to mitochondrial fission and mitophagy (mitochondrial autophagy). Without proper clearance of dysfunctional mitochondria, CECs may become taxed to the point of cell death, leading to the higher amounts of dropout observed. Additional studies are indicated in this area, particularly given recent prospective randomized clinical trial data documenting increased graft failure rates and greater CEC loss three years following DSAEK surgery using diabetic graft tissues [9,10].

In addition to these functional changes, T2DM CECs have several structural matrix properties that may explain their tendency to tear during surgical preparation. Mechanical peel testing during controlled separation of donor corneal EDM tissues from stroma showed higher mean values for elastic peel tension (TE), average delamination tension (TD), and maximum tension (TMAX) in advanced diabetic donor corneas compared to non-diabetic donor corneas [27]. The region being peeled, between Descemet membrane and the stroma, is known as the interfacial matrix. Alterations occurring in diabetes to this region may be responsible for the higher mechanical peel test results, but further research is required. Using transmission electron microscopy (TEM), Rehany et al. found there were abnormal 120-nm wide-spaced collagen fibril bundles within both Descemet membrane and the stroma of noninsulin dependent diabetic patients (*n* = 16) compared to nondiabetic controls (*n* = 16) [28]. The authors hypothesized that the wide spacing was due to excessive glycosylation products. Together, the mechanical abnormalities and structural abnormalities in Descemet membrane, the interfacial matrix, and the stroma may explain the increased risk of tearing diabetic EDM grafts during donor tissue preparation for Descemet membrane endothelial keratoplasty [8].

## 3. Glaucoma

Abnormalities of the ECM have been implicated in some models of glaucomatous optic neuropathy. Cellular and extracellular matrix (ECM) interactions contribute to the resistance at the trabecular meshwork to aqueous outflow [29]. Increased deposition or impaired remodeling of extracellular matrix, changes in actin fiber contractility and arrangement, and regulatory derangements of cell adhesion all appear to play a role in the pathophysiology of glaucoma [30]. Vascular signaling also appears to play a role, as recent studies of nitric oxide and cyclic guanosine monophosphate signaling show alterations in vascular tone in many anterior segment structures as well as improved outflow facility [31,32]. This has been adapted for therapeutic use [33,34]. However, it is still unclear whether this signaling pathway involves any mediation or interaction with extracellular matrix, therefore the rest of this sub-section will focus on pathways that do.

### 3.1. Trabecular Meshwork

The ECM is a vital component of all three segments of the trabecular meshwork: the corneoscleral, uveoscleral, and juxtacanalicular layers. The trabecular meshwork ECM is comprised of numerous glycosaminoglycans and proteoglycans, collagens, elastic fibrils, basement membrane, and matrix proteins [29]. In the corneoscleral and uveoscleral layers, the trabecular meshwork cells wrap around these components to form the trabecular beams, between which are relatively large intertrabecular pores (Figure 4) [30]. In the juxtacanalicular layer, which is the site of highest resistance to aqueous outflow, trabecular meshwork cells have a more interwoven and irregular spatial relationship with ECM fibrils [35].

The components of the trabecular meshwork ECM are dynamic in that their expression or function can be induced by interaction with other components via bidirectional signaling, or as responses to environmental stimuli, such as mechanical stretch [30,36,37,38,39,40]. As one example, the matrix metalloproteinases (MMPs) can be induced by mechanical stretch, glucocorticoid steroids such as dexamethasone, laser trabeculoplasty, and the inflammatory cytokines TNF-*α* and TGF-*β*, leading to ECM remodeling and subsequent alterations to outflow facility [41,42,43]. Processes that impede MMP function and ECM remodeling may therefore decrease outflow facility [36,38,40,44,45]. Similarly, the expression and/or induction of numerous other proteoglycans and matricellular proteins, such as tenascin C, thrombospondin-1 and -2, SPARC (secreted protein, acidic and rich in cysteine), connective tissue growth factor (CTGF), fibronectin, various integrins, and periostin, have also been associated with changes in intraocular pressure [30,41,46,47,48,49,50,51,52,53,54,55,56,57,58,59,60,61,62,63,64,65,66,67]. These mechanisms are not limited to ECM turnover, and also include modulation of such processes as cellular contraction, adhesion and migration, proliferation, and phagocytosis [60,68,69,70,71] (Figure 5). It has been suggested that cell–matrix interactions like these are mechanisms by which the trabecular meshwork can regulate intraocular pressure homeostasis [30], and elements in these interactive pathways, such as Rho kinase inhibitors, have shown promise as therapeutic targets [71,72,73].

### 3.2. Optic Nerve

Optic nerve head (ONH) remodeling in the glaucomatous excavation process also appears to be modulated by cellular and ECM interactions. The lamina cribrosa is a porous support structure for retinal ganglion cell axons passing through the scleral canal and out of the eye. The lamina cribrosa contains three cell types, including astrocytes, lamina cribrosa cells, and microglia [74,75,76], all of which are interspersed with the ECM. The ECM of the lamina cribrosa is comprised of proteoglycans as well as the relatively stiff collagens and relatively flexible elastin [77]. The last two mediate the distensibility of the lamina cribrosa, which overall becomes less mechanically compliant with age that is further accentuated in glaucoma [37,77,78]. In glaucomatous eyes, astrocytes at the prelaminar region have relatively enhanced expression of collagen type IV mRNA, while lamina cribrosa astrocytes have de novo expression of elastin mRNA. Animal models of glaucoma have more elastotic fibers at the lamina cribrosa compared to models of nerve transection or fellow eyes, suggesting that the increased elastin synthesis is a response to increased intraocular pressure [74]. Stress–strain models and analyses further support the notion that ECM deposition and dysregulation of ECM remodeling are eventual responses to mechanical stretch [37,79,80]. TGF-*β*1 and MMPs appear to have a role in mediating this pathway, and many other genes, including elastin, collagens IV, VI, VIII and IX, thrombospondin, perlecan, and lysl oxidase, show increased expression in response to 24 h cyclical stretch [42,75,81]. It has been postulated that the intraocular pressure-induced changes in the laminar ECM may even impede axonal nutrition despite stable laminar capillary flow [79].

### 3.3. Conjunctival and Tenon’s Layer

Subconjunctival fibrosis is also a process involving overproduction of ECM and is of particular interest for subconjunctival glaucoma surgeries such as trabeculectomy. After a filtration procedure, fibroblasts of Tenon’s layer (human Tenon’s fibroblasts, HTFs) may be chemoattracted to the surgical area by such factors as fibronectin in the aqueous [82]. after which there is increased ECM synthesis and collagen contraction [82,83,84]. Component proteins of the subconjunctival ECM include fibronectin and collagen type I, which are further induced by TGF-*β*1 [85]. One matricellular protein, SPARC, has been noted to be significantly increased in HTFs after exposure to TGF-*β*1 or TGF-*β*2, and relatively more SPARC is found in scarred blebs compared to normal Tenon’s. Additionally, SPARC-null knockout mice have HTFs that do not respond to TGF-*β*1, and filtration surgery in these mice functioned longer and with more expansive blebs than compared to wild type mice [86,87]. Another matricellular protein, CTGF, seems to promote bleb scarring and has been found to be overexpressed in filtration blebs [88,89]. Subconjunctival injection of a CTGF antibody after filtration surgery in rabbits also led to relatively larger blebs and lower intraocular pressure [90]. These matricellular proteins may hold promise as therapeutic targets for enhancing filtration surgery efficacy.

## 4. Vitreous

The vitreous humor is unique in that it is comprised almost entirely (>98%) of water but has various ECM components that give its gel-like consistency at birth and liquefies with age. Multiple blinding disorders can result from pathologic changes at the vitreoretinal interface. Normal vitreous is nearly completely acellular except for some macrophage-like hyalocytes [91]. The vitreous contains a network of glycosaminoglycans (GAGs), primarily hyaluronan, that supports a scaffold of collagen fibrils allowing a swelling osmotic gradient to inflate the gel [92].

The most prevalent form of collagen found in the vitreous is type II, which is secreted as procollagen prior to cleavage. Alternative splicing of exon 2 pre-mRNA can yield two forms: if exon 2 is expressed it is called procollagen IIA (which more prevalent in the vitreous); if exon 2 is excluded it is procollagen IIB [93,94]. Other collagens that are less prevalent but have a relatively prominent role in the vitreous include types V/XI, [95] IX, [93] XVIII, and VI [96].

Orientation of collagen fibrils varies in different regions of the eye. The central vitreous fibrils tend to course parallel in an anterior-posterior direction as opposed to the vitreous base where the fibrils insert directly into the internal limiting membrane (ILM) perpendicularly [97,98]. The precise mechanisms for adhesion of the vitreous to the ILM are not fully understood but differs at the vitreous base compared to the rest of the eye.

### 4.1. Age-Induced ECM Changes in the Vitreous Causing Retinal Detachment

Aging results in vitreous liquefaction and weakening of vitreoretinal adhesion that is associated with loss of type IX collagen and its chondroitin sulfate side-chains, and a four-fold increase in ‘sticky’ type II collagen predisposing to fibril fusion [99,100]. Opticin on the surface of cortical vitreous collagen fibrils may bind heparan sulphate proteoglycan chains on the ILM, including type XVIII collagen, which can also mediate vitreoretinal adhesion [101,102]. The role of opticin in angiogenesis from proliferative diabetic retinopathy (PDR) is discussed further in Section 5.1 but it is noteworthy that complete posterior vitreous detachment is protective against PDR as the collagenenous scaffold network for neovascularization is no longer present.

Vitreoretinal adhesion is critical in formation of retinal breaks and ensuing rhegmatogenous retinal detachment during the process of posterior vitreous detachment [103,104]. Lattice degeneration increases risk for retinal tears as there is increased vitreoretinal adhesion in these lesions with overlying vitreous liquefaction as well as alterations in the ILM, absence of basement membranes over the lattice, and increased presence of astrocytes [105,106].

### 4.2. Posteriorly Inserted Vitreous Base

The vitreous base can migrate posteriorly with advanced age that could be due to synthesis of new collagen by retinal cells [107]. Posteriorly inserted vitreous base (PIVB), generally defined as a wider than average vitreous base that straddles the ora serrata, has been observed in human donor eyes [108,109,110]. We defined vitreous base as posteriorly inserted if the posterior hyaloid membrane could not be elevated during pars plana vitrectomy anterior to the vortex veins, which approximates the equator of the eye that averages a distance of 7.6 mm posterior to the ora serrata in human donor eyes (Figure 6). PIVB can present challenges to eyes undergoing vitrectomy due to the increased number retinal tears (average 3.1) pre-operatively, high incidence of new breaks occurring during vitrectomy (30%), and increased risk for proliferative vitreoretinopathy needing re-operation [111]. Primary scleral buckle can be used for some of these cases but if vitrectomy is employed for a retinal detachment with PIVB, use of a wider buckle, meticulous shaving of the vitreous base, 360 degree laser, longer-acting tamponade agents, and potentially removal of crystalline lens [112] may help reduce rates of re-operation and vision loss [113].

## 5. Retina and Retinal Pigment Epithelium (RPE)

In the retina, ECM is organized into the interphotoreceptor matrix (IPM) and the retinal ECM (RECM) [4]. The IPM (Figure 7) represents the meshwork occupying the subretinal space between the photoreceptor cells and the retinal pigmented epithelium (RPE), and is comprised of a unique array of glycoproteins, while the RECM represents ECM outside the IPM. Structurally, ECM is found in basement membranes including the inner limiting membrane, the vasculature and Bruch’s membrane (BM) [1]. The major source of RECM are the Müller cells, intraretinal and migrating glial cells [114] while most of the IPM components, of which hyaluronan (HA) forms the basic scaffold, are synthesized by either the RPE or photoreceptors [115].

Within a given tissue, the ECM is a milieu in constant evolution and could show variation in its composition and organization over time [4]. For instance, age-related posterior vitreous detachment [116]. A well-known physiologic phenomenon, and ILM increasing in thickness and stiffness [117]. Both ECM-driven processes are mediated by changes in retinal cellular differentiation, migration, and adhesion [1,118,119,120]. Other physiologic processes related to ECM functionality and breakdown include tissue wound healing, innate immune defense, and angiogenesis [121,122,123,124].

### 5.1. Retinal Endothelial Cells and Angiogenesis

The pathophysiology of angiogenesis with relation to ECM are complex and not fully understood yet. Retinal angiogenesis is vitally important in age-related macular degeneration (AMD; discussed in Section 5.3) and blinding retinal vascular diseases such as proliferative diabetic retinopathy (PDR) and retinal vein occlusion, where preretinal neovascularization could result in massive pre-retinal hemorrhage, contractile fibrovascular membranes and tractional retinal detachment. Before detailing the role of ECM in vessel formation, it is useful to understand two concepts of angiogenesis: sprouting and intussusceptive [125]. In sprouting angiogenesis, new vessels are formed after an initial endothelial “tip cell” degrades the pre-existing vessel basement membrane, migrates into the surrounding ECM, proliferates and directs the remaining “stalk cells” in the formation of a cord [125]. This process is dependent on ECM-based growth factor signaling including vascular endothelial growth factor (VEGF) [125]. Intussusceptive angiogenesis, which is the basis for preretinal neovascularization in PDR [126] is different as the newly formed vessel emerges from the splitting of pre-existing blood vessels [125].

The ECM plays a critical role in most, if not all, aspects of vascular biology. ECM vasculogenic functionality includes the following: (1) supporting key signaling events in endothelial cell adhesion, proliferation and survival, (2) providing a scaffold and organizational cues for endothelial cells, (3) control and orchestration of the endothelial cells’ cytoskeleton via integrin-dependent signal transduction pathways, (4) tubulogenesis and three-dimensional remodeling of endothelial cell sheets, and (5) vessel maturation and stabilization [127].

Endothelial cell morphogenesis follows a programmed stepwise chain of events that starts with basement membrane breakdown, followed by cellular migration and proliferation, and ending with lumen formation and stabilization [128,129]. The concept of “fire and ice” was introduced to describe the role of endothelial ECM in vascular biosynthesis, remodeling, morphogenesis, and stabilization [127]. In fact, there is an ECM-based signaling balance which dictates when endothelial cells are activated or stabilized [127]. The process of activation will eventually lead to basement membrane degradation, cellular invasion, proliferation, migration and lumen formation. Both collagen (via interaction with *α*1*β*1, *α*2*β*1 integrins), fibrin, and fibronectin (via interaction with *α*5*β*1 and *α*V*β*3 integrins) are essential for the “fire” chain of events, in conjunction with VEGF (Figure 8) [127].

During new blood vessel sprouting from pre-existing vasculature, membrane-type matrix metalloproteinases (MT-MMPs) are important regulators of cellular invasion into adjacent collagen or fibrin matrices due to their role in degrading ECM proteins at the cell surface–ECM interface [128,130,131,132]. MMPs themselves are regulated by tissue inhibitors of metalloproteinases (TIMPs), including TIMP-1, TIMP-2, TIMP-3, and TIMP-4, and it is the balance between MMPs and TIMPs that controls membrane degradation [132,133]. For instance, the “tip cell” utilizes MT1-MMP to degrade the surrounding ECM [134]. TIMP-2 and TIMP-3 are then subsequently produced when “stalk cells” contact pericytes in an attempt to halt the MT1-MMP induced ECM degradation [135].

Endothelial cells must then structurally adhere to the adjacent ECM in order to migrate [136]. The process of adhesion to ECM is mediated via specific surface integrins and ensures endothelial cell proliferation, survival and directional motility [137,138,139,140]. Endothelial cell proliferation is potentiated by activation of the p44/p42 (Erk1/Erk2) mitogen-activated protein kinase (MAPK) signal transduction pathway, which is itself activated by adhesion to fibronectin, a key ECM component [141,142,143]. Endothelial cell survival is also ensured by their adhesion to the ECM, which is a powerful regulator of Fas-induced apoptosis [144]. When attached to the ECM, endothelial cells are protected from apoptosis [144]. ECM modulates Fas-mediated apoptosis by altering the expression of Fas and c-Flip, an endogenous antagonist of caspase-8, which is a proteolytic enzyme involved in programmed cell death [144,145,146].

ECM also regulates endothelial cell morphogenesis and contractility. The matrix-integrin-cytoskeletal signaling axis results in both sprouting (= cord formation) and luminal vacuolization, which ultimately connect endothelial cells tubular structure together [127,128]. Collagen I promotes shape changes that lead to precapillary cord formation witnessed during angiogenesis [147,148,149]. Endothelial cells cytoskeletal contractility, which drive cord assembly, is potentiated by collagen I interaction with (1) integrins *α*1*β*1 and *α*2*β*1, which suppress the cAMP-dependent protein kinase A [150], and (2) integrin *β*1, which activates Src kinase and the GTPase Rho [151].

The interaction between laminin and each integrin *α*1*β*1, *α*2*β*1, *α*3*β*1 and *α*6*β*1 in conjunction with TGF-*β* are essential for “ice” chain of events which result in endothelial cell differentiation and stabilization via the cessation of cellular proliferation and related morphogenic sequelae [125,127,146].

It is important to note that most of the angiogenic pathways described above are highly conserved and are not unique to human species and their retina. Specifically, many genes necessary for animal multicellularity including fibronectin [152], cadherin [153], integrins [154], extracellular matrix domain [155,156], and VEGF machinery [157,158] code for ancient highly conserved proteins. Interestingly, many of these proteins were conserved all the way to the ‘‘Urmetazoan” [156] and multiple phyla [159]. These demonstrate the importance of these canonical pathways in vascular biology and, consequently, mammalian and non-mammalian species development [160,161,162].

### 5.2. Angiogenesis and Fibrosis in Proliferative Diabetic Retinopathy

As mentioned above, there is a substantial role for ECM in controlling intussusceptive angiogenesis, which is the basis for pre-retinal neovascularization seen in PDR [125,126]. This pathologic venous-based angiogenesis is an attempt to re-vascularize the ischemic retinal areas. In such cases, the cortical vitreous serves as a structural scaffold for pre-retinal neovascularization, specifically using vitreous ECM as a primary substrate for their formation [125,126]. Initially, the provisional vascular matrix was found to contain fibronectin and vitronectin [163], followed by collagen types I and III, which are deposited by fibroblast-like cells [125,164].

Interestingly, the vitreous normally hosts an anti-angiogenic milieu thanks to ECM components such as opticin, thrombospondins and endostatin (a fragment of type XVIII collagen) [125], which are well-known anti-angiogenic mediators [165,166,167]. In fact, opticin, which belongs to a family of ECM leucine-rich repeat proteoglycans [168], is a powerful dose-dependent inhibitor of preretinal neovascularization [169]. Hence, there must be a pro-angiogenic signaling shift on the collagen fibril surfaces within the vitreous ECM of PDR eyes. Opticin and other anti-angiogenic mediators could be affected by the enzymatic degradation of MMPs (1, 2, 3, 7, 8 and 9) and ADAMTS-4 and -5 (a family of extracellular protease enzymes, short for multi-domain extracellular protease enzyme) [170].

Research in PDR brought up the idea of an ‘angiofibrotic switch’ as a shift in balance between VEGF and CTGF mediating angiogenesis and fibrosis, respectively [171,172]. The basis for this work was done primarily on clinical grading and aqueous fluid extracted from patients. However, membranes removed from patients in a reverse translational, randomized controlled trial using VEGF inhibition for end-stage diabetic fibrovascular membranes demonstrated that VEGF and CTGF were not significantly different between intervention groups despite suppression of VEGF fluid levels in those that received bevacizumab (Figure 9) [173,174]. CTGF levels in the vitreous and aqueous were also unchanged in controls and those receiving bevacizumab, but a fair number of these patients had severely fibrotic, end-stage membranes where a change in CTGF would not have been as likely [174]. This study as well as others found that eyes receiving bevacizumab may have higher levels of apoptosis [174,175], supporting the notion that VEGF inhibition induces contraction of blood vessels rather than obliteration of them [176,177]. Endothelial-to-mesenchymal transition may be involved in diabetic membrane formation as there is evidence that endothelin-1, a potential vasoconstrictor that promotes fibrosis, is present at higher levels in diabetic compared to non-diabetic epiretinal membranes [178].

### 5.3. Choroidal Neovascularization, Autosomal Dominant Radial Drusen and AMD

In degenerative retinal diseases (whether acquired or inherited), there can be a tipping point at which the degenerative process is accelerated, leading to phenotypic manifestations of the disease [4]. This could be represented by the loss of a critical mass of retinal ECM [4]. Evidence supporting this hypothesis stems from numerous studies conducted in age-related macular degeneration (AMD), the most common cause of irreversible blindness in the developed world [179]. Drusen, which are extracellular lipid filled deposits between the RPE and the choriocapillaris, are the earliest hallmarks of AMD and tend to form over ECM areas with low density or absent choriocapillaris [180,181,182]. Additionally, drusen in AMD do not express collagen type IV in contrast to drusen to patients with genetic mutation in epidermal growth factor–containing fibrillin-like extracellular matrix protein 1 (EFEMP-1) causing autosomal dominant radial drusen (ADRD, aka Malattia Leventinese and Doyne Honeycomb Retinal Dystrophy) (Figure 10) [183]. Furthermore, the density of drusen correlates well with the density of “ghost” choriocapillaris vessels that is independent of RPE cell height [181,184,185].

The initial insult driving pathologic changes in AMD is not well understood. Early pathophysiological changes can be localized to the choriocapillaris, where an abundance of membrane attack complex (MAC) resulting in aberrant complement activation has been reported [182,186,187,188,189]. MAC based complement injury to choriocapillaries could be irreversible that leads to uncontrolled angiogenic drive and the formation of choroidal neovascular membranes (CNV) [185] and geographic atrophy [190]. Acute complement injuries have been associated with higher levels of MMP-3 and -9 [182].

The pathogenesis of CNV in AMD is complex and several interconnected pathways including genetic predisposition, oxidative stress, inflammatory/immune mechanism, and angiogenesis play a role [191,192]. Monogenic inherited retinal diseases directly affecting extracellular matrix such as ADRD from *EFEMP-1* mutation results in early geographic atrophy and CNV, which is responsive to anti-VEGF therapy [193]. Despite the overwhelming success of anti-VEGF intravitreal injections (IVI) in treating active CNV due to exudative AMD [194,195,196], there is still a subset of incomplete respondents (~15%) who have persistent sub-retinal fluid (with or without intra-retinal fluid) despite chronic continued treatment [197,198,199]. Interestingly, some incomplete responders may initially show a good response to anti-VEGF IVI but then become treatment resistant and lose significant vision over time [200,201,202].

The mechanism of resistance to anti-VEGF IVI treatment is unknown. Tachyphylaxis was previously proposed as a possible explanation, especially in those eyes that show initial improvement [203,204,205]. There may also be a special role of MMP-mediated immune response in angiogenesis and CNV formation [206,207,208,209]. Of note, MMPs are essential in the degradation of ECM and basement membrane [210,211] and thus are crucial in tissue remodeling and repair [212,213]. Their major targets are elastin, fibrinogen, gelatin and various types of collagen molecules, including I, IV and V [214].

Most targets of MMP-9 are structural components of BM, which forms a major angiogenic barrier to CNV-based insult in exudative AMD [183,215,216]. There is mounting evidence of the causal relationship between MMPs, BM pathological remodeling, and CNV in AMD [215,217,218,219,220]. MMPs were reported to be increasingly expressed in pathologically stressed tissue, such as BM of eyes with AMD [221,222]. Breakdown of BM structural molecules (specifically elastin and collagen IV) allows for the migration of endothelial cells during angiogenesis [16,223,224]. Both MMP-9 enzymatic activity and the incidence of exudative AMD increase with age, suggesting a correlative risk [183,215,216]. MMP-9 is expressed in choroidal macrophages [225] and has also been found near BM the margins of CNV membranes [207]. Additionally, MMP-9 was found to increase the RPE VEGF levels by decreasing levels of pigment epithelium-derived factor (PEDF), which is the main antagonist of VEGF in the RPE [226,227,228]. Both VEGF and PEDF are highly expressed in AMD and their interplay serves as a mediator in the development of CNV [229,230,231,232,233]. A significant reduction in CNV incidence and severity was reported in MMP-9 knockout mice artificially subjected to laser injury [234,235]. In addition, inhibition of MMP-9 was experimentally found to block CNV development [235,236]. In humans, exogenous MMP-9 upregulated the gene expression of VEGF in human RPE cells [237,238]. In their study, Liutkevicien et al. showed a significant association between MMP-9 specific single nucleotide polymorphism and the incidence of AMD at a younger age (<65-year-old) [238]. Chau et al. also reported three folds higher plasma [239] and aqueous humour [240] levels of MMP-9 in AMD patients compared to healthy controls.

Furthermore, MMP-9 is known to interact with TIMP-3, mutations of which cause Sorsby fundus dystrophy in which CNV invariably develops patients by their 4th-5th decade of life [241,242,243]. Additionally, AMD eyes with marked choroidal thinning due to geographic atrophy have been reported to have a marked increase in TIMP-3 activity [244,245]. This is could then hinder the normal choroidal physiological angiogenic repair and contribute to the observed choroidal thinning [244]. In addition, there is a genetic association of the *MMP-9* locus with exudative AMD, which was found in the International AMD Genetics Consortium in a large genome wide association study [245] and independently confirmed recently in an Iowa cohort of patients with AMD [246].

## 6. Bruch’s Membrane and Choroid

Bruch’s membrane is a complex, multilayered extracellular matrix compartment. It is comprised of two layers with salient features of fibrillar collagen surrounding a central layer of elastin and related molecules (Figure 11). While the basal laminae of the RPE and choriocapillaris are sometimes considered the inner and outer boundaries of Bruch’s membrane, increasingly investigators find it useful to consider these separately [247]. Both the structure and pathology of Bruch’s membrane are reminiscent of the arterial wall in atherosclerotic disease [248].

Bruch’s membrane itself has numerous functions: it serves as a barrier to abnormal neovascularization from the choroid, occupying the interface between the abundant vasculature of the choriocapillaris and the avascular outer retina. It contains both structural proteins and matricellular proteins. An abbreviated list of constituents of this unusual ECM compartment includes collagens I, III, IV, V, VI, VIII, fibrillin-1, fibulins 3 and 5, TIMP3, MMPs, and antiangiogenic effectors [249,250,251,252,253,254]. Several ECM constituents of Bruch’s membrane become greatly reduced during aging and macular disease, concomitant with the accumulation of lipidic debris [255,256], including the anti-angiogenic matricellular protein thrombospondin [257]. Age-related structural and biochemical changes in Bruch’s membrane are mirrored by an age-related decrease in hydraulic conductivity [258]. The elastic layer of Bruch’s membrane further becomes fragmented in AMD, with a loss of elastin integrity quantifiable both by histochemical staining and ultrastructural appearance [259]. Elastin degradation results in the liberation of elastin derived peptides (EDPs) which have been found to be elevated in the circulation of patients with neovascular AMD [260]. Liberated elastin fragments activate choroidal endothelial cells to migrate toward the source of the peptides, which ECs detect using a heterotrimeric cell surface elastin receptor.

The loss of elastin in Bruch’s membrane (whether due to increased metalloproteinase activity, macrophage extravasation, increased brittleness due to calcification, and/or other events) has the potential to both create a physical opening for growing vascular tubes as well as signaling choroidal endothelial cell migration. Moreover, human primary choroidal ECs stimulated with elastin fragments increase MMP-9 expression, potentially promoting further loss of Bruch’s membrane elastin and further amplification of angiogenesis [261]. The reader is referred to several reviews for more information (Figure 11) [216,262,263].

Outside of Bruch’s membrane, the choroidal ECM has been relatively understudied. The choroid is comprised of several cell types with their own basal laminae, including endothelial cells (which differ in composition between cells at different positions in the vascular tree and between the choriocapillaris and RPE) [253], pericytes/smooth muscle cells surrounding capillaries and large vessels respectively, and Schwann cells [225]. Other abundant cells such as melanocytes and especially fibroblasts contribute to ECM synthesis. The choroidal stroma, which occupies the space between and around vascular lumens, includes abundant fibrillar collagens with a ground substance containing a complex array of glycosaminoglycans including heparan sulfate and chondroitin sulfate proteoglycans [264]. Choroidal thinning, observed in normal aging and in geographic atrophy, is characterized by persistence of collagen fibrils and loss of ground substance, with a shift in the balance of serine protease inhibitors and metalloproteinase inhibitors as described above [244]. Other clinically meaningful aspects of choroidal thickness, such as the changes that occur in pachychoroid spectrum diseases like central serous chorioretinopathy, remain to be elucidated.

## Figures and Tables

**Figure 1 cells-10-00687-f001:**
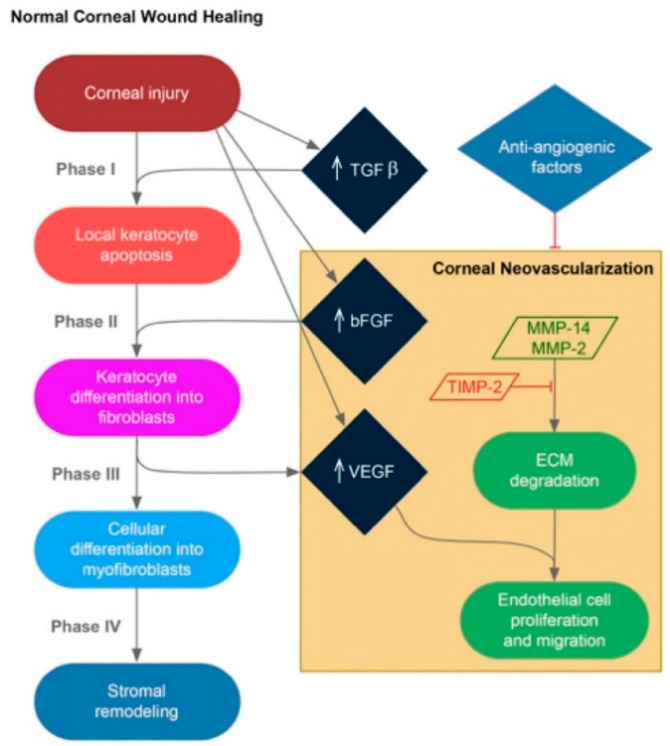
Corneal stromal wound healing. Stromal wound healing in the cornea is mediated by signaling of transforming growth factor-beta (TGF-*β*), matrix metalloproteinases (MMPs), and a balance between pro-angiogenic and anti-angiogenic factors. In some cases, corneal neovascularization can occur.

**Figure 2 cells-10-00687-f002:**
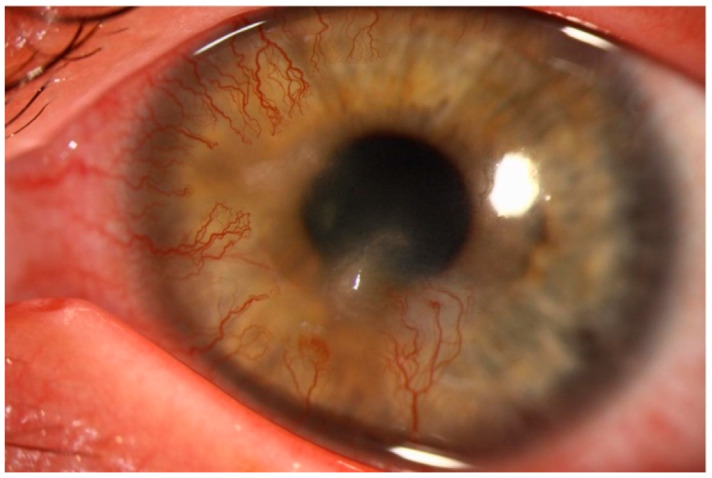
Corneal neovascularization with stromal scarring secondary to atopic keratoconjunctivitis.

**Figure 3 cells-10-00687-f003:**
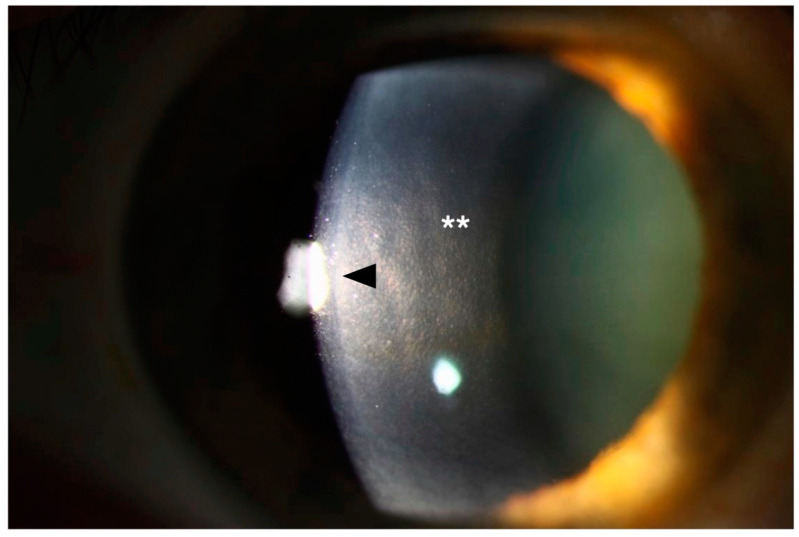
Extracellular fibrosis (arrowhead) and dense guttae (**) in the posterior cornea secondary to Fuchs endothelial corneal dystrophy.

**Figure 4 cells-10-00687-f004:**
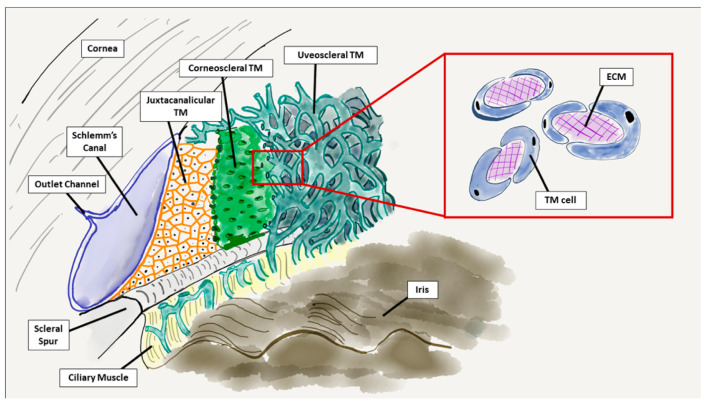
Configuration of the trabecular meshwork. In both the corneoscleral and uveoscleral layers, the trabecular meshwork cells wrap around a core of extracellular matrix components. Note the increasingly large intertrabecular pores between the trabecular meshwork beams in the deeper layers. In the juxtacanalicular layer, the extracellular matrix and the trabecular meshwork cells have a more irregular and interwoven spatial relationship.

**Figure 5 cells-10-00687-f005:**
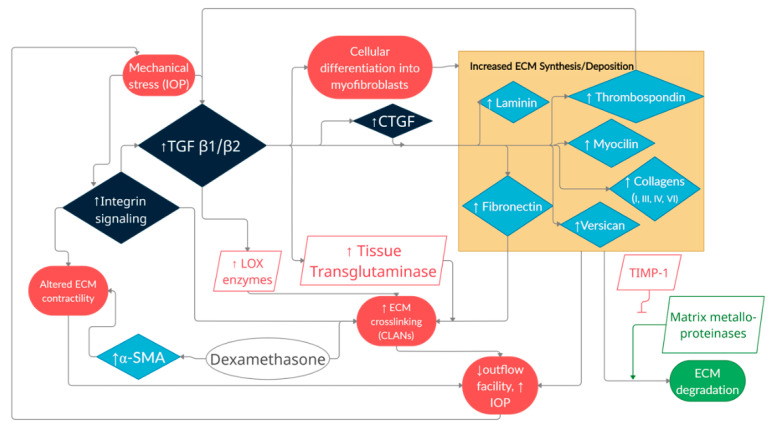
Overview of cellular and extracellular matrix interactions in glaucomatous tissue remodeling. The interplay of numerous factors, including environmental stressors, enzymatic reactions, growth factors, glycoproteins and proteoglycans as well as cytoskeletal elements all contribute to a feedback loop where outflow facility is disturbed. Red circles indicate negative situational change. Green circle indicates positive situational change. Dark blue diamonds highlight critical signaling factors. Light blue diamonds indicate extracellular matrix glycoproteins and proteoglycans. Red rhombuses indicate negative catalysts. Green rhombus indicates positive catalysts. White circle indicates exogenous factors. ECM: Extracellular matrix; TGF: Transforming growth factor; CTGF: Connective tissue growth factor; *α*-SMA: Alpha smooth muscle actin; CLANs: Cross-linked actin networks; TIMP-1: Tissue inhibitors of metalloproteinases 1.

**Figure 6 cells-10-00687-f006:**
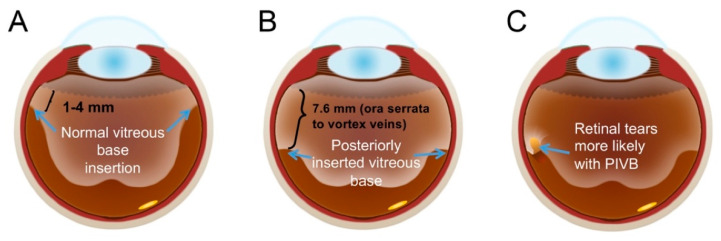
Schematic of vitreous base inserting into the retina. At the pars plana normally (**A**), when it is posteriorly inserted (**B**), or at posterior the equator, averaging 7.6 mm posterior to the ora serrata which predisposes to more retinal tears (**C**). Adapted from Sohn et al. [111].

**Figure 7 cells-10-00687-f007:**
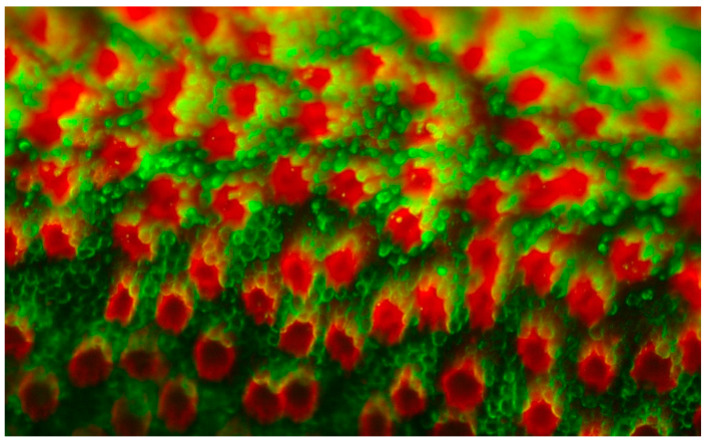
Insoluble interphotoreceptor matrix glycoproteins, the gene products for *IMPG1* and *IMPG2* are distributed in domains surrounding rod and cone photoreceptors. The relative distributions of cone matrix sheaths labeled with peanut agglutinin (red) is depicted compared to rod outer segments labeled with anti-rhodopsin (green).

**Figure 8 cells-10-00687-f008:**
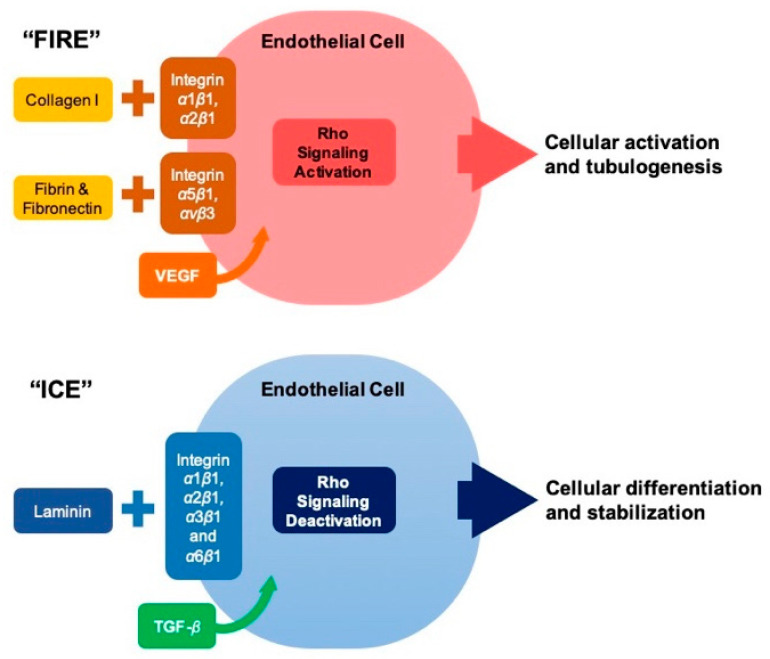
Extracellular matrix mediated endothelial morphogenesis. The diagram illustrates the concept of “fire and ice” representing balance of extracellular matrix-based signaling which dictates endothelial cellular activation and tubulogenesis with endothelial cell stabilization. The complex process is mediated by the interaction between extracellular matrix components (including collagen I, fibrin, fibronectin and laminin) and various integrins. Abbreviations: VEGF: vascular endothelial growth factor; TGF-*β*: transforming growth factor beta.

**Figure 9 cells-10-00687-f009:**
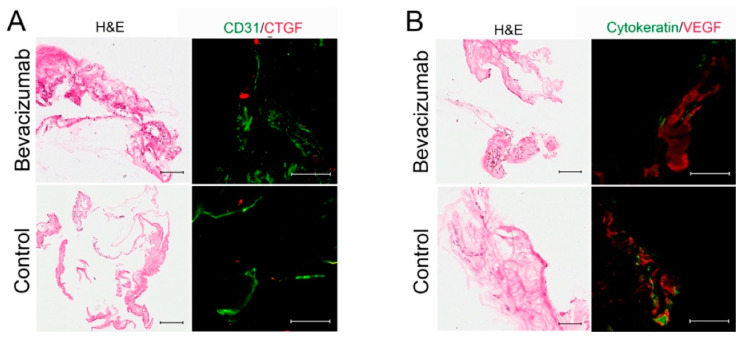
Representative hematoxylin and eosin (H&E) and immunofluorescence images from four patients’ membranes in a randomized controlled trial. Co-labeling of antibodies for (**A**) CD31 (Green)-CTGF (Red) and (**B**) cytokeratin (Green)-VEGF (Red). Note the H&E-stained sections do not correspond precisely to the cytokeratin-labeled sections. While intravitreal bevacizumab did not significantly decrease CTGF (A-top panels) or VEGF (B-top panels) expression in membranes compared to sham group, VEGF was still expressed in membranes of eyes given bevacizumab (**B**, right panels). Scale bar = 100 μm. Abbreviations: CTGF: connective Tissue Growth Factor; VEGF: vascular endothelial growth factor. Adapted from Jiao et al. [174].

**Figure 10 cells-10-00687-f010:**
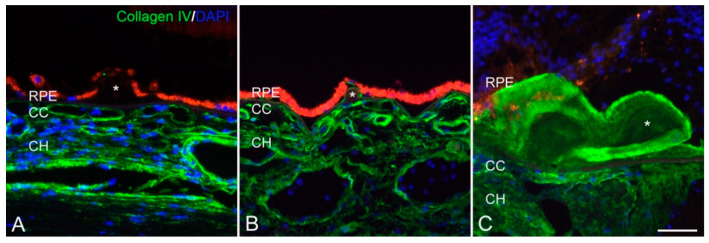
Anti-collagen IV labeling in human donor eyes. Drusen (asterisks) associated with aging do not show labeling with antibodies directed against collagen type IV (**A**–**C**). Laminae within the autosomal dominant radial drusen are immunoreactive with anti-collagen IV antibodies (green fluorescence). Sections were also labeled with DAPI (blue nuclear fluorescence) and were exposed in the rhodamine channel (red autofluorescence of the RPE). Scalebar = 50 μm. Adapted from Sohn et al. [183].

**Figure 11 cells-10-00687-f011:**
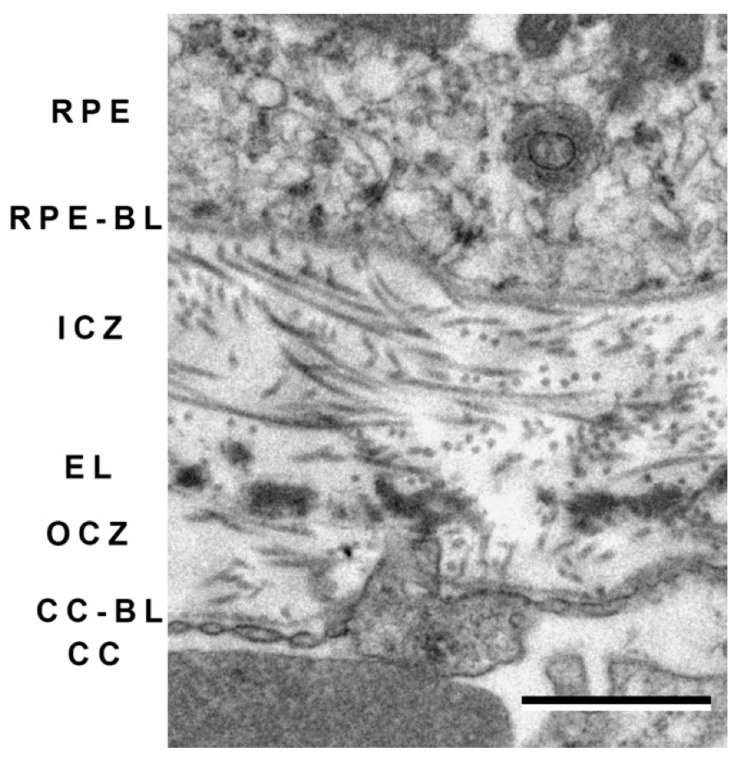
Transmission electron micrograph depicting the layers of Bruch’s membrane from a human eye. Both the basal laminae of the RPE and choriocapillaris (RPE-BL and CC-BL) are depicted, in addition to inner collagenous zone (ICZ) and outer collagenous zone (OCZ), occupied by fibrillar collagens, as well as the elastic lamina (EL), evident by its thick electron dense bundles. Scale bar = 1 μm.

## Data Availability

Not applicable.

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
