# Peer review of "Cell–Matrix Interactions in the Eye: From Cornea to Choroid"

_cells, 2021, doi:10.3390/cells10030687_

Round 1

Reviewer 1 Report

The authors have constructed a well-organized review, followed by a primary description of a newly modified experimental methodology.  I highly recommend separating these out into two separate publications.  In my opinion the review article is acceptable for publication in present form, but the primary description of new explant methodology requires additional experimentation.

With regard to the review article: The writing is clear and effective.  Figures provide appropriate illustrations of the main concepts.  Figures including Figure 10 are exceptional.  Some figures are reproductions from previous publications; in these cases, please confirm that the copywrite holders have granted permission.

Please add references for line 55-57 and 100-103, referencing pathological settings for corneal opacification

Please add references for line 69-71, describing age related EC death.  Does the 0.4% figure reflect a particular species?  Is loss constant or increasing rate with increasing age?  When endothelial cells die, do the remaining endothelial cells expand their surface area to prevent formation of frank breaches of endothelial continuity, or do whole vessel segments occlude?  Decreased levels of cell-cell junction proteins and NA+/K+ ATPase does not necessarily need to correlate with endothelial cell death (eg line 77); even without cell death the existing ECs can dynamically change their expression of key proteins.

To make Figure 3 more accessible to a wide readership, please expand the figure legend and/or add dotted line or arrow to more clearly describe.  Consider pairing with an image of a healthy non-occluded cornea with similar lighting and magnification for comparison.

Line 188, typographical error, replace period with comma.

Line 195, please include the citation for Aldrich et al.  It is not necessary (line 197) to provide the manufacturer of the Seahorse; this manuscript is not a primary report.

Consider adding a schematic to illustrate lines 238-243.

Section 5.1, is a carefully cited and well-written description of canonical angiogenesis.  However, please add a discussion of how these canonical cascades may be unique to the retina, versus conserved with other endothelial vascular beds.  For example, Are the ECM components different? Reliance on Rho?  

Line 519, reference 169, Mullins and colleagues did not measure RPE cell height, but density.

Figure 9, please add scale bar.

With regard to the description of porcine explants:

Figure 11, it is unclear to have two superimposed timelines (-48h to 96 hours and Day 0-Day 6-7).  Please consolidate into a single timeline.

Does tubular outgrowth continue to improve after 7 days in vitro?

Does vascular pruning (ECM sleeves or vascular ghosts) take place in this explant system, and if so, when? 

Please carefully assess the immunostaining of CD31 in Figure 13A.  What is the explanation for the diffuse halo of CD31 that extends out and beyond the edges of GSL-1 positive EC cell membranes? Higher resolution higher magnification insets would be helpful. Instead of a stereo pair, consider simply incorporating orthogonal views.

Please provide further characterization of the pericyte component in the explants. The statement in line 759 that few, if any NG2 positive cells are present is inconsistent with Figure 13C, which shows many NG2 positive cells.  Please add additional immunostaining for alternative pericyte markers.

Figure 12 and Figure 13B-C shows some regions to contain well-formed tubules, but also contains several regions that are out of focus or only show GSL-1+ blebs that presumably correlate to tubule cross sections. This depiction should be improved by presenting a maximal intensity projection of a confocal Z-stack.

Author Response

Reviewer 1

Please add references for line 55-57 and 100-103, referencing pathological settings for corneal opacification

As requested, we have added the comprehensive manuscript by Tshionyi et al (PMID 22030600) in support of these lines.

Please add references for line 69-71, describing age related EC death. Does the 0.4% figure reflect a particular species? Is loss constant or increasing rate with increasing age? When endothelial/cells die, do the remaining endothelial cells expand their surface area to prevent formation of frank breaches of endothelial continuity, or do whole vessel segments occlude? Decreased levels of cell-cell junction proteins and NA+/K+ ATPase does not necessarily need to correlate with endothelial cell death (eg line77); even without cell death the existing ECs can dynamically change their expression of key proteins.

We thank the reviewer for these comments. As requested, we have added a reference on endothelial cell aging and death by Bourne and Hodge (PMID 9071233). To address all of the reviewer’s queries, we have replaced the previous sentence on lines 70-71 with an updated sentence that addresses these points (“As a result, human corneal endothelial cell density decreases at an average rate of approximately 0.6% per year in normal corneas throughout adult life, with gradual increases in polymegethism and pleomorphism to maintain a continuous endothelial cell layer.”)

With respect to the reviewer’s comments about protein level dynamic shifts, we agree that expression changes may occur with cell dropout in remaining cells. However, we do not know for certain if this is true, the duration of such changes, or whether such changes may compensate for the degree of cell loss. We do know, however, that corneal edema results from endothelial cell loss, and therefore believe that even if pump and junction protein levels change to compensate that these changes are not suffiecient to maintain proper cell function. Therefore, we respectfully decline to make further changes to lines referenced by the reviewer on this point.

To make Figure 3 more accessible to a wide readership, please expand the figure legend and/or add dotted line or arrow to more clearly describe. Consider pairing with an image of a healthy non-occluded cornea with similar lighting and magnification for comparison.

We thank the reviewer for their comment. We have added an arrowhead to denote fibrosis and an asterisk to highlight corneal guttae, and have revised the figure legend as follows: “Figure 3. Extracellular fibrosis (arrowhead) and dense guttae (asterisks) in the posterior cornea secondary to Fuchs endothelial corneal dystrophy.”

Line 188, typographical error, replace period with comma.

We appreciate the review’s attention to this error, which is now corrected.

Line 195, please include the citation for Aldrich et al. It is not necessary (line 197) to provide the manufacturer of the Seahorse; this manuscript is not a primary report.

We have made both changes as requested by the reviewer.

Consider adding a schematic to illustrate lines 238-243.

We have added an original schematic and a figure legend to demonstrate this.

Section 5.1, is a carefully cited and well-written description of canonical angiogenesis.  However, please add a discussion of how these canonical cascades may be unique to the retina, versus conserved with other endothelial vascular beds.  For example, Are the ECM components different? Reliance on Rho? 

We have added the following paragraph to the end of Section 5.1:

‘It is important to note that most of the angiogenic pathways described above are highly conserved, and are not unique to human species and their retina. Specifically, many genes necessary for animal multicellularity including fibronectin[153], cadherin[154], integrins[155], extracellular matrix domain[156, 157], and VEGF ma-chinery[158, 159] code for ancient highly conserved proteins. Interestingly, many of these proteins were conserved all the way to the ‘‘Urmetazoan”[157] and multiple phyla[160]. These demonstrate the importance of these canonical pathways in vascular biology and, consequently, mammalian and non-mammalian species development [161-163].’

Line 519, reference 169, Mullins and colleagues did not measure RPE cell height, but density.

The reviewer is correct that RPE cell height was not presented in this paper but we have added two other references that discuss this.

Figure 9, please add scale bar.

A scale bar has been added (=50 micrometers)

As the comments to the description of porcine explants:

We appreciate the reviewer’s comments regarding this section. We have removed this section from this review and will submit it as a separate manuscript.

Reviewer 2 Report

The manuscript entitled “Cell-matrix interactions in the eye: from cornea to choroid” by Pouw and colleagues, describes the role of extracellular matrix in the physiological and pathological states of the eye in a front to back approach: starting from ocular surface to posterior chamber of the eye.

The manuscript is well written and pleasant to read. It is appreciable the text layout based, for each section, on the eye’s anatomy, with a brief introduction, and related associated pathologies. One exception is the section dedicated to glaucoma, which sounds a bit off-key. It is, indeed, tricky to identify a specific etiopathogenesis for glaucoma, so I would suggest moving glaucoma under a section “Intraocular pressure maintenance and Glaucoma”.

Speaking about the ocular surface, the authors forgave to mention two pathologies in which the remodelling of the extracellular matrix is essential, specifically keratoconus and pterygium. It is not mandatory, but I believe that a subsection would complete the manuscript.

Figures are adequated.

The abstract is not engaging or intriguing. It should be rewritten from the second sentence to the end since the last sentences are more or less a copy of the introduction’s second paragraph. In my opinion, the authors should address the abstract content to the cell-matrix interactions in a more general and comprehensive point of view, avoiding repetition of the introduction.

References are adequated and exhaustive

Author Response

Reviewer 2

Speaking about the ocular surface, the authors forgave to mention two pathologies in which the remodelling of the extracellular matrixis essential, specifically keratoconus and pterygium. It is not mandatory, but I believe that a subsection would complete the manuscript.

We thank the reviewer for this comment. We have chosen to explore representative corneal pathologies to highlight the importance of cell-matrix intractions on corneal function and vision. In the interest of achieving timely publication, we will defer coverage of additional corneal pathologies that feature perturbations to cell-matrix interactions such as keratoconus, anterior basement membrane dystrophy, corneal transplant rejection, and microbial keratitis.

The abstract is not engaging or intriguing. It should be rewritten from the second sentence to the end since the last sentences are more or less a copy of the introduction’s second paragraph. In my opinion, the authors should address the abstract content to the cell-matrix interactions in a more general and comprehensive point of view, avoiding repetition of the introduction.

We appreciate this comment and wish we had enough space to include details of all of the interactions and pathways discussed in the review but the 200 word limit is challenging. As suggested, we have highlighted some of the pathways discussed by adding this sentence to the middle of the abstract:

‘A variety of pathways and key factors related to the ECM in the eye are discussed, including, but not limited to, those related to transforming growth factor-b, vascular endothelial growth factor, basic-fibroblastic growth factor, connective tissue growth factor, matrix metalloproteinases (in-cluding MMP-2, MMP-9, and MMP-14), collagen IV, fibronectin, elastin, canonical signaling, integrins, and endothelial cell morphogenesis consisting of cellular activation-tubulogenesis and cellular dif-ferentiation-stabilization.’

Round 2

Reviewer 1 Report

The manuscript provides an excellent review of the literature.  The new/modified figures significantly enhance the manuscript. Thank you for your attentive revisions in response to previous requests.  At this time the only additional comment is that the text is difficult to read in Figure 5, and would benefit from being enlarged.